# Innovative Reports on the Effects of Anabolic Androgenic Steroid Abuse—How to Lose Your Mind for the Love of Sport

**DOI:** 10.3390/medicina59081439

**Published:** 2023-08-08

**Authors:** Michał Stojko, Jakub Nocoń, Patrycja Piłat, Gabriela Szpila, Joanna Smolarczyk, Karol Żmudka, Martyna Moll, Michał Hawranek

**Affiliations:** 1Student’s Scientific Society, III Department of Cardiology, Faculty of Medical Sciences in Zabrze, Medical University of Silesia, 40-055 Katowice, Poland; 2Student’s Scientific Society, Department of Psychiatry, Faculty of Medical Sciences in Zabrze, Medical University of Silesia, 40-055 Katowice, Poland; 3Department of Psychiatry, Faculty of Medical Sciences in Zabrze, Medical University of Silesia, 50-055 Katowice, Poland; 4III Department of Cardiology, Faculty of Medical Sciences in Zabrze, Medical University of Silesia, 40-055 Katowice, Poland

**Keywords:** anabolic-androgenic steroids, innovations, testosterone, nervous system, symptoms

## Abstract

Anabolic-androgenic steroids (anabolic-androgenic steroids, AAS) are testosterone-derived compounds whose popularity and use are constantly growing. Chronic use of AAS leads to many hormonal and metabolic disorders in the human body, which often lead to permanent health damage. Changes affect the following systems: cardiovascular, musculoskeletal, reproductive, digestive, and nervous. We decided to collect the existing knowledge in the literature and enrich it with the latest research reports in the field of degenerative effects of AAS on the nervous system. The work aimed to increase public awareness of the dangers and consequences of AAS use and improve it with the latest research on the neurodegenerative effects of AAS. We hope that our work will contribute to raising public awareness and reducing the use of AAS.

## 1. Introduction

Testosterone is the most commonly used AAS, and its anabolic effect consists of retaining electrolytes in the body, increasing synthesis, and reducing protein catabolism. It supports the growth of skeletal muscles, causing a positive nitrogen balance, and supports bone mineralization. Exogenous testosterone taken in healthy men results in the inhibition of the secretion of endogenous testosterone and spermatogenesis, which may lead to infertility. In women, it has an antagonistic effect on estrogens, and it can lead to masculinization with hypertrophy of the clitoris, hirsutism, irregular menstrual cycles, and modulation in the tone of the voice. Changes in the reproductive system in men may partially disappear after discontinuation of use, but unfortunately in women, many of them may be irreversible [1,2]. The literature also describes numerous AAS side effects associated with damage to the cardiovascular, immune, nervous, and hematological systems and impaired liver function while more recent reports concern neurodegenerative changes.

### 1.1. Mechanism of Action of Anabolic-Androgenic Steroids

Anabolic-androgenic steroids (AAS) are a group of hormones that include testosterone and many of its synthetic derivatives [3]. The biological effects of SAAs and their metabolites are mainly mediated by intracellular receptors present in the brain and reproductive system as well as many non-reproductive tissues such as bone, skeletal muscle, liver, kidney, and adipocytes [4,5]. Brain receptors for sex hormones are widespread in the brain, particularly affecting regions such as the brainstem, hypothalamus, amygdala, striatum, hippocampus, and cortex [4,5]. Receptors for sex hormones are widespread in the brain, especially in such regions as the brainstem, hypothalamus, amygdala, striatum, hippocampus, and cerebral cortex. The known modifications of the testosterone molecule concern the alkylation of the 17α carbon structure, the aim of which is to obtain derivatives with more anabolic and less androgenic effects than the basic testosterone molecule, and esterification of the 17β-hydroxyl group by carboxylic acids, which increases the activity of the obtained compound and prolongs its action by obtaining better lipophilicity. The anabolic effect is related to the retention of nitrogen which results in an increase in its bioavailability to tissues and a decrease in the activity of glucocorticosteroids which reduces catabolic reactions. In connection with the aforementioned modifications, some testosterone derivatives, for example, Nandrolone (Nandrolone contains a substituted hydrogen atom at the C19 methyl group of testosterone which creates a new asymmetric center at C10 which may be responsible for the increased anabolic to androgenic ratio), belong to the group of testosterone derivatives with increased anabolic to androgenic effects which contribute to its increased benefits in therapy and lower reproductive side effects [4,6,7].

### 1.2. Prevalence of AAS in the Population

AAS supplementation is often associated with the temptation to achieve a significant increase in lean muscle mass in a short time and increase physical capacity [8]. These effects are highly desirable among athletes, which is confirmed by the fact that the use of AAS in sports was observed as early as 1950, and gained the greatest popularity in the 1980s [9,10]. Currently, the use of doping is widespread not only among people who train professionally (which is additionally strictly controlled by The World Anti-Doping Agency and International Olympic Committees [11]) but also among amateurs and young people starting their adventure with various sports disciplines. The motivation to start using such support is the desire to increase muscle mass, strength [12,13], and self-confidence [14]. The pressure caused by social media has a particular impact here, in which the ideal figure associated with a flat belly, low percentage of body fat, and a visibly muscular body is widely promoted, or even normalized [15,16,17]. Artificially created models of a woman and a man in our society become, even subconsciously, a main goal to achieve, which can cause stress, frustration, and anxiety. Such behavior may have a destabilizing effect on everyday functioning and be a motivation to reach for boosters [18,19].

AAS and SARMs (Selective Androgen Receptor Modulators) have become a serious public health problem [20,21]. The ease of obtaining and insufficient knowledge about the substances often lead to the development of numerous side effects. Survey studies show that over 50% of people abusing AAS obtain them via the Internet, 16% are from local sources such as friends or training partners, and as many as 15% of respondents have AAS prescribed by a doctor [22]. In the latest analysis of the global prevalence of AAS, it was estimated that 3.3% of the population had once taken AAS without health indications, of which 6.4% were men and 1.6% were women [23]. The Crime Survey for England and Wales (CSEW) indicated that 31,000 people aged 16–59 are most likely to abuse AAS [24]; however, new research indicates that this statistic is underestimated, and the actual use of doping is ten times higher [25].

### 1.3. Strategies to Prevent Misuse of AAS

The primary strategies for preventing the misuse of AAS involve restricting their legal distribution. As mentioned earlier, the Olympic Committees banned the doping use of AAS and included them on the list of banned substances in 1967. Many countries have registered AAS as prescription drugs, which carries a penalty of deprivation of liberty in case of their illegal distribution (for example, in Poland there is a penalty of deprivation of liberty for up to two years on the basis of Article 124 of the Pharmaceutical Law Act of 2001) [26].

An important factor in preventing the misuse of AAS is to raise public awareness of the harmful effects of these substances through research and analysis, which this paper is also aimed at.

## 2. Clinical Use of AAS

Before we discuss in detail the most important side effects of taking AAS, it is worth mentioning their clinical use. In addition to the androgenic preparations commonly prescribed by endocrinologists for the treatment of hypogonadism in men, AAS can be used in the treatment of severe cachectic conditions which are a result of the Acquired Immune Deficiency Syndrome (AIDS), cancer, burns, trauma, liver failure, kidney failure, and anemia. The positive effects of therapy with the use of AAS for the above-mentioned conditions mainly concern the problem of weight loss resulting from an inadequate ratio of food supply to changes in energy expenditure, malabsorption as well as hormonal and metabolic disorders [27]. The main therapeutic goal of using anabolic androgenic steroids (AAS) apart from treating hypogonadism and supporting puberty is to utilize the anabolic effects of these substances to improve the patient’s condition and prevent muscle loss in individuals with chronic diseases [4].

## 3. Known Systemic Side Effects of AAS

### 3.1. Cardiovascular System

The cardiovascular system is one of the most sensitive systems in the matter of the harmful effects of AAS. The use of these compounds increases vascular resistance and blood pressure, the profile of pro-inflammatory biomarkers, and the activity of the sympathetic nervous system, which is associated with changes in serum lipoproteins and has a toxic effect on the myocardium [28,29]. Cardiovascular health problems attributed to AAS use include cardiomyopathies [30], left ventricular hypertrophy, hypertension [31], left ventricular systolic dysfunction, dyslipidemias [32,33], myocardial infarction [34], class IV heart failure NYHA scale (New York Heart Association) [35], arterial thrombosis, pulmonary embolism [36], coronary atherosclerosis [37] and sudden cardiac death (SCD) [1].

In 1995, Melchert and Welder proposed four hypothetical models to explain how AAS causes cardiovascular side effects. They noted an increase in serum lipoprotein concentrations and thus a predisposition to the development of atherosclerosis, increased thrombocyte aggregation and thus thrombus formation, nitric oxide release and vasoconstriction, cardiac toxicity with accompanying fibrosis, and changes in microcirculation [38]. An important impact of chronic AAS use is the dysfunction of tonic autonomic regulation of the heart [39], as reflected in an experimental study from 2006. It demonstrated that rats exposed to AAS exhibited impaired parasympathetic modulation and heart rate variability [40]. There are also reports that the inflammatory process may induce damage to the cardiac muscle in individuals abusing AAS, as observed in a 2011 experimental study. In the group of mice treated with AAS, a strong cytokine response was observed compared to the control group, leading to the conclusion that TNF-α may play a role in cardiac damage [41]. The effect of chronic AAS use on lipid metabolism was emphasized in a study involving men with hypogonadism undergoing testosterone replacement therapy, which resulted in a reduction in high-density lipoprotein cholesterol (HDL) levels in the serum [42]. Subsequent studies conducted in the previously mentioned direction showed significantly elevated levels of low-density cholesterol (LDL), which is important in the development of atherosclerosis [42,43,44].

There are also reports that abuse of anabolic steroids can promote the growth of heart tissue, and this phenomenon is associated with ventricular remodeling, cardiomyopathy, myocardial infarction, and SCD. This may explain how AAS overuse presumably leads to death from myocardial infarction without thrombosis or coronary atherosclerosis [37,45]. In histopathological studies, changes such as myocardial fibrosis and necrosis were observed in athletes abusing AAS. Those may be responsible for atrioventricular conduction disorders and be the basis for the occurrence of fatal arrhythmias and SCD [46,47,48,49]. In the assessment of the myocardium, it is very important that, regardless of the use of anabolic-androgenic steroids, the increased dimensions of the heart cavities, wall thicknesses, and mass of the left ventricle are the consequences of high-intensity exercise training and are part of the physiological remodeling of the heart known as an “athlete’s heart”. It is true that slight cardiac fibrosis may arise physiologically as a result of long-term endurance training [41]. However, it has been shown that the use of AAS in combination with endurance training changes the physiological remodeling of the athlete’s heart into pathophysiological myocardial hypertrophy, which significantly increases the risk of life-threatening adverse events [36,41].

### 3.2. The Musculoskeletal System

The main temptation to start abusing anabolic androgenic steroids is to increase lean muscle mass and improve the body’s performance in a relatively short time. The anabolic effect in skeletal muscles exerted by AAS is obtained due to the interaction with the androgen receptor (AR), which affects the induction of hypertrophy of muscle fibers type I (slow fibers) and type II (fast fibers), as well as by an increase in the number of myocytes and capillaries per one fiber [50]. These effects are mediated by the stimulation of muscle protein synthesis using the GH/IGF-1 axis (Insulin-like growth factor 1) and mesenchymal stem cells [51].

The effect of testosterone on bone tissue is manifested in both the support of periosteum formation, transverse growth, and increased bone density. This explains the larger cross-sectional size of male bones compared to female bones. This action is mediated by the androgen receptor along with estrogens converted from testosterone as a result of the RANKL/RANK/OPG system (Receptor Activator for Nuclear Factor κB Ligand/Receptor Activator of NF-kB/Osteoprotegerin, NF-κB/Osteoprotegerin) by stimulating osteoblasts and suppressing osteoclasts [52]. SAA given to athletes in childhood and adolescence causes acceleration of bone maturation, so such supplementation may consequently lead to short stature by epiphyseal atresia [53,54].

Muscle damage usually occurs as a result of AAS overuse and independently strong and long-lasting tissue load. If they become chronic, negative health effects will often be felt even after the discontinuation of sport and doping [55].

It is worth noting that during intensive use of AAS, rhabdomyolysis was also observed [56]. The increase in strength and muscle mass obtained with the use of supportive compounds is presumably not accompanied by an equal increase in tendon strength [57]. However, this impression is based only on single observations, and a follow-up study is not available [55]. In 2008, an experimental study was conducted on rats, monitoring the activity of MMP-2 (Matrix metallopeptidase-2) in selected tendons, as well as the influence of AAS on the concentration of this enzyme. The authors divided the rat population into sedentary and exercise-trained groups, along with their counterparts receiving additional AAS supplementation. In the exercise-trained population without AAS supplementation, an increase in MMP-2 activity was observed, reflecting active tissue remodeling. However, in the AAS-supplemented groups, not only a decrease in MMP-2 activity was noticed, but also inhibition of the exercise training effect. The authors caution that currently, it is not clear how AAS modulates the concentration and activation of MMP-2. Nevertheless, the impact on tendons is likely a reflection of a delayed anabolic response of this tissue compared to the highly vascularized and androgen-responsive skeletal muscles [58]. In a survey of 2552 former American football players, no association was found between AAS abuse and tendon or muscle injuries. On the other hand, in the same experiment, people abusing adjuvants had a higher incidence of injuries to the ligaments and meniscuses of the knee, elbow, spine, and ankle joint. However, the authors of the study warn against causal conclusions, because athletes often took AAS after injury in order to regenerate faster, and the study itself does not provide information on when these compounds were taken [59].

### 3.3. Reproductive System

Anabolic-androgenic steroid doses taken by people using them often exceed 50–100 times the physiological values of testosterone produced by the testicles, which is associated with many side effects in the male reproductive system [60].

The basic mechanism that disturbs active spermatogenesis in these men is the AAS causing negative feedback on the hypothalamic–pituitary–testis axis, which results in the inhibition of the pulsatile secretion of GnRH (gonadoliberin), then a decrease in the secretion of FSH (follicle stimulating hormone), LH (luteotropic hormone), and a reduction in the production and release of testosterone by Leydig cells. The lack of hormonal stimulation causes atrophic changes in Leydig cells, which manifests by reducing the volume of the cytoplasm and condensing the nucleus [61]. The role of FSH and testosterone is to prevent apoptosis of reproductive cells and seminiferous tubules covered with Sertoli cells; therefore, with persistently reduced concentrations of these hormones, testicular volumes decrease significantly. In addition, the tubules occupy about 95% of the volume of the testicles, and their disappearance causes the testicles to shrink [62,63]. Such hormonal changes may lead to azoospermia, oligospermia, hypogonadotropic hypogonadism, and a reduction in the percentage of morphologically normal spermatozoa. The recovery period of the hypothalamic–pituitary–testis axis usually takes up to four months and depends on the patient’s age and the amount of doses taken. Younger men have the ability to restore axis function more quickly than older men [60,64]. Nonetheless, long-term use may be associated with the irreversibility of changes, which has been shown in many studies over the past years [65,66,67].

Gynecomastia, i.e., enlargement of the mammary gland in men, sometimes with adipose tissue, is one of the most common complications of AAS use, as it stands for as much as 50% of reported side effects. It may occur during the cycle due to developed hypogonadotropic hypogonadism or administration of hCG (chorionic gonadotropin), which is used to prevent the detection of AAS use [68]. Fatigue and depression are common after discontinuation of AAS, as a consequence of the sudden drop in androgenic hormone levels in the blood directly related to the withdrawal of exogenous androgens as well as the reduction of endogenous androgen production [68,69,70].

The use of AAS by women carries the risk of certain changes in the reproductive system as well. Women using AAS are also exposed to a number of side effects, which at over-physiological doses may cause permanent changes in the reproductive system. It should be noted that the mechanism of disruption of the hypothalamic–pituitary–gonadal axis works on the same principle as described above in men. Negative feedback occurs, which consequently causes a decrease in the secretion of GnRH by the hypothalamus, leading to a decrease in the secretion of LH and FSH by the pituitary which ultimately causes changes in female sexual characteristics and the functioning of the reproductive system. The irreversible nature of the changes is associated with increased susceptibility to the masculinizing effects of AAS [69]. The literature describes the association of SAA use with ovarian follicle immaturity and uterine, clitorial, vaginal, and mammary gland hypertrophy [70,71]. The most common side effects are hormonal disorders, at 61% [72]. Virilization, as a result of masculinization by AAS, is associated with voice pitch change, hirsutism, hair loss, acne, increased libido, and hypertrophy of the clitoris similar to congenital hypoadrenocorticism [72,73,74]. Women using the AAS chronically may experience fertility disorders that result from a disruption of the menstrual cycle and are caused by a lack of ovulation [74,75].

### 3.4. Digestive System

The liver is sensitive to the AAS due to its large number of androgen receptors and the fact that it is the main detoxification organ of the body [76,77,78]. Hepatotoxicity can manifest itself on a variety of backgrounds, from elevated liver enzymes to life-threatening liver failure. Secondary liver damage caused by the use of AAS is classified as a type of drug-induced liver damage [77] and may include diseases such as cholestasis, fatty liver, and chronic vascular damage and can lead to the development of cancer. AAS administered orally with modification of the 17-α-alkyl group are better tolerated than esterified AAS administered in injections, but they tend to cause more side effects in terms of hepatotoxicity than parenteral preparations [76]. It is suspected that AAS has a hepatotoxic effect by activating Kupffer cells, resulting in the production of inflammatory cytokines and collagen deposition, increased oxidative stress, mitochondrial degradation, and irregular growth of hepatocytes due to activation of the androgen receptor [79]. In 2021, an in vitro study was conducted and showed that the use of over-physiological doses of synthetic AAS-nandrolone caused the inhibition of complexes I and III of the respiratory chain and, as a result, the accumulation of reactive oxygen species [80]. However, other studies on animal models have described reduced activity of glutathione and enzymes that neutralize free radicals, e.g., superoxide dismutase and catalase [81,82,83]. Disturbed dynamics of the respiratory chain affect the potential of the mitochondrial membrane, which results in reduced energy supply to hepatocytes, leading to their malfunction and necrosis [83]. Excessive collagen deposition in the liver parenchyma is caused by AAS-induced infiltrative changes. Those concern the infiltration of inflammatory cells, which together with Kupfer cells induce inflammation by releasing many proinflammatory factors, e.g., NF-κB, TNFα, IL-1B. These factors lead to the activation of hepatic stellate cells, resulting in excessive collagen synthesis and deposition [84]. However, attention should be paid to the differences in AAS hepatotoxicity depending on the route of administration, pre-existing liver diseases, and individual susceptibility.

### 3.5. Skin Changes

Skin changes associated with AAS appear in the form of papular acne, which is caused by hypertrophy and an increase in the number of sebaceous glands, and thus increased secretion of sebum [65,66]. SAAs also cause increased production of skin surface lipids. These changes are caused by increased sebum synthesis through direct binding to the androgen receptor in sebocytes and indirectly through induction of nuclear peroxisome proliferator receptors (PPARs), which are involved in sebaceous cell growth and differentiation [66]. Another skin symptom of chronic abuse of AAS is androgenic alopecia, also known as male pattern baldness or genetic hair loss. The mechanism of AAS-induced baldness involves the shortening of the hair growth cycle, which normally lasts from 2 to 6 years but can be reduced to a few months in individuals using these substances. As a consequence, the hair becomes thinner and shorter over time. This process is accompanied by the miniaturization of hair follicles, leading to their eventual disappearance. AAS acts on the hair follicles, causing newly regrown hair to become progressively thinner and weaker. It is important to note that androgenic alopecia can occur in men who do not use AAS and is influenced by genetic predispositions, but the use of AAS increases the likelihood of hair loss [85,86].

### 3.6. Nervous System

The latest research describing the effect of AAS on the human body mainly covers the effect on the central nervous system; therefore, we considered it appropriate to devote a separate part of the paper to this topic in order to highlight the importance of this research.

## 4. Potential Methods to Mitigate the Negative Effects of AAS

The use of AAS carries the risk of many adverse effects, which have already been described in this study. However, for individuals using AAS for therapeutic purposes under a doctor’s recommendation or illegally using these substances on their own initiative, methods to reduce their side effects are sought. The most important factor in minimizing the side effects of AAS is to follow the doctor’s advice, take the prescribed doses, and assume proper preparations. In case of any concerning symptoms, it is important to consult a doctor, as only a qualified professional can properly assess our health status. Another factor that can help limit adverse effects is the use of legal preparations (pharmaceutical products prescribed by a doctor) and avoiding products of unknown origin or from the black market. Regular blood tests conducted to monitor hormone levels can detect abnormalities at an early stage and allow for prompt medical intervention. It is important to remember that the use of AAS for therapeutic purposes involves the use of minimal doses, and any individual increases in doses by the patient can contribute to the occurrence of adverse effects [87,88]. There are also extreme cases of combating the side effects of chronic AAS use. A noteworthy example is a case report describing a 27-year-old woman who experienced voice changes due to hyperandrogenism after 6 weeks of AAS use. Despite discontinuing the use of these substances, the patient experienced irreversible vocal fold thickening. The patient was qualified for surgical vocal fold narrowing. Furthermore, the surgical procedure resulted in minimal improvement, and after 10 years, the patient returned to the hospital for further diagnostics [89].

## 5. Innovative Reports on the Effects of AAS

### 5.1. Effect on the Central Nervous System

The latest reports on the use of over-physiological doses of AAS relate to the significant acceleration of brain aging processes, cognitive dysfunction, and psychosocial disorders. This is related to the easy access of AAS to CNS cells due to their ability to penetrate the blood–brain barrier and the rich supply of AAS receptors, which are concentrated in the brainstem, amygdala, hypothalamus, striatum, cerebral cortex, and hippocampus. Worth noting is that the negative effects on the CNS are not only related to direct exposure but are also associated with disorders of other organs secondary to the use of AAS.

### 5.2. Psychosocial System

Another observed consequence of the pathological effect of AAS is the frequent occurrence of psychological problems among users and the increased tendency to abuse other substances [88]. Over the years, many case reports and surveys have been presented, which indicate that the use of AAS may cause aggressive behavior [90,91,92]. Initially, people using AAS noticed increased irritability, agitation, and anger [93]. With long-term use of these compounds, loss of inhibitions, mood swings, suspiciousness, argumentativeness, impulsiveness, and aggressive behaviors [94] have been recorded, sometimes resulting in assaults on public property, self-harm, domestic violence [90], child abuse [95], suicides [96,97], attempted homicides, and murders [98]. An area that is poorly researched among the population using AAS is psychopathy, i.e., a personality disorder characterized by disturbances in the emotional sphere, lack of empathy, and antisocial behavior [99]. In 2022, a study was published that involved 492 adult male bodybuilders. The conclusions were consistent with previous studies, as it was proven that the intake of AAS is associated with a higher probability of developing psychopathic features and engaging in risky sexual behavior. Moreover, non-users who had already been considering this kind of supplementation demonstrated higher tendencies in psychopathic traits, anger problems, emotional instability, and depressive symptoms compared to ones who never considered the use of AAS. However, the authors warn against drawing cause-and-effect conclusions and recommend continuing research on this issue [100].

There are many reports presenting cases of brutal behavior of men acting under the influence of AAS. Several publications describe men with no history of violence or criminal behavior prior to anabolic steroid use who committed or attempted murder while on doping supplementation [91,100]. A review article mentioned six cases of AAS-induced violence, including three homicides and three violent assaults. The perpetrators were described as aggressive and irritable. The authors reported that all six convicts returned to a normal mental state within two months of discontinuation of AAS [101]. In another experiment, four placebo-controlled trials were conducted in the 1990s and early 2000s, with participants using weekly doses of testosterone in the range of 500–1440 mg [102,103,104]. A total of 109 men were evaluated, five of whom developed hypomanic or manic syndromes during the administration of AAS, while none of the above appeared in participants receiving a placebo [105]. Neuroimaging studies have revealed a probable biological basis for AAS-induced impulsive behavior. The amygdala is a symmetrical structure located in the anterior temporal lobe that plays a key role in regulating emotions and controlling aggressive behavior. Moreover, it is structurally and functionally connected to parts of the frontal cortex involved in executive functions and the so-called top–down cognitive control, which physiologically limits aggression [106]. The right amygdala has been observed to be enlarged in chronic AAS users, and functional magnetic resonance imaging (fMRI) results have shown its reduced connectivity at rest with the frontal cortex [107], which may impair top–down cognitive control and regulation of amygdala activity and reactivity [105]. Amygdala enlargement is associated with increased aggressive behavior in humans [108], and decreased resting-state connectivity between the amygdala and the frontal cortex during the fMRI is associated with an increased risk of violence [109]. Another study conducted in rats showed that in castrated adult males, the volume of the corticomedial area of the amygdala was underestimated, and this effect was abolished with the use of exogenous testosterone [110].

### 5.3. Effect on the Neurodegenerative Processes

In 2021, a study was published to determine the brain age gap (BAG, brain age gap) on a group of 229 men, in which 130 participants were confirmed to have prolonged use of AAS based on magnetic resonance imaging (MRI). The BAG is the difference between the chronological age and the predicted age of the brain. As the chronological age, a reference to the MRI scans of a group of 1838 healthy men aged 18–92 was used. For all participants, BAG age was estimated based on features of the whole brain and its subregions. A statistically significant increase in the predicted age of the brain compared to the control group was found for the frontal, temporal, crest, cingulate, and occipital areas in the group of patients using AAS for more than 10 years in high doses.

MRI studies in earlier years yielded similarly disturbing results. In 2017, a study was published in which a group of weightlifters using AAS was compared to a group of non-takers. The study showed a bilaterally significantly thinner cerebral cortex in chronic AAS users.

The literature shows that over-physiological doses of AAS can induce the process of apoptosis of nerve cells. Over the years 2006–2019, exemplary models of CNS damage developed on the basis of in vitro tests were described, the results of which coincide with the results of imaging tests.

The model of neurotoxicity proposed in 2019 is based on the aggregation of beta-amyloid and tau protein, resulting from the increase of oxidative stress in CNS neurons.

Accelerated accumulation of Tau protein and beta-amyloid is associated with the influence of AAS on the expression and function of enzymes involved in their synthesis and elimination, e.g., amyloidogenic β- and γ-secretases and amyloid precursor protein. This process is associated with an increased risk of Alzheimer’s and other neurodegenerative diseases. However, it is proven that physiological testosterone concentrations of 10 nM and 17β-estradiol may prevent neurodegenerative processes. Testosterone can be aromatized to the neuroprotective molecule 17β-estradiol, which protects against the development of Alzheimer’s disease, etc. [111,112,113].

In 2013, the effect of AAS on cognitive functions was demonstrated in a rat model. Spatial memory and learning were assessed using the Morris water maze. A deterioration in coping with the maze obstacle was found in the AAS-treated group, which is related to the decreased expression of choline acetyltransferase in the basal forebrain induced by AAS treatment. When analyzed using immunoenzymatic techniques, a decrease in the level and expression of NGF (Nerve Growth Factor) was detected in the basal forebrain [114].

In 2009, the effect of physiological doses of testosterone on dopaminergic neurons was proven. The study was designed to verify the influence of androgens on the development of Parkinson’s disease. The project used dopaminergic neurons due to the origin of Parkinsonism. Based on the results, it was shown that androgens can cause dopaminergic cell death through a cascade starting with the impairment of mitochondrial function and increasing oxidative stress. The increased amount of free oxygen radicals activates the caspase-3-dependent cleavage of protein kinase C, leading to DNA fragmentation and neuronal apoptosis [115].

Another suggested mechanism leading to neurodegeneration is the effect of testosterone in over-physiological doses on intracellular calcium metabolism. Testosterone induces intracellular Ca^2+^ oscillations in the cytosol and cell nucleus, which are important mediators of further transformations, including neurite growth. In several cell models, it has been shown that the intracellular increase of calcium ions can contribute to the induction of apoptosis. The use of testosterone in low concentrations had no effect on cell viability, but with the gradual increase in testosterone concentration, the viability of nerve cells decreased [116].

## 6. Conclusions

The above-presented literature review underlines how little we know about the pathophysiological processes and effects of AAS on the neurodegenerative processes and the overall effects that they have on the nervous system. Despite the fact that most countries prohibit the distribution and use of AAS, their abuse is still a serious problem for public health. In this review, we particularly elaborated the impact of these compounds on neurodegenerative processes and the increased tendency towards aggressive behaviors and psychotic states among AAS users. By utilizing brain age prediction techniques based on MRI measurements, clear evidence was gathered to confirm the influence of excessive AAS use on brain aging. We also referenced studies showing a correlation between AAS administration and the enlargement of the amygdala in both humans and animals, which directly affects the development of aggressive behaviors. Another important fact is that AAS usage is becoming increasingly popular among young individuals, and acquiring these substances is not difficult. We believe that this trend will persist until the issue is brought to light and relevant information is widely disseminated. Introducing informational campaigns regarding the side effects of doping and updating medical knowledge and awareness among healthcare professionals about AAS-induced adverse events are crucial.

Despite the numerous known pathologies caused by AAS abuse, there are still many unknown mechanisms, making further research in this field necessary.

## Data Availability

Not applicable.

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
