# Peer review of "Innovative Reports on the Effects of Anabolic Androgenic Steroid Abuse—How to Lose Your Mind for the Love of Sport"

_medicina, 2023, doi:10.3390/medicina59081439_

Round 1
Reviewer 1 Report
Firstly, I commend you on this comprehensive study of anabolic androgenic steroid (AAS) abuse. The potential side effects, mechanism of action, and prevalence are well presented. The narrative is well structured and the use of references is commendable. However, here are some points you might want to consider:
1. Abstract:
1. The abstract seems to be a bit heavy on content. Consider summarizing the findings more concisely to improve readability.
2. There's no clear mention of your objective or purpose. Include a statement that explains the aim of your review or perspective, e.g., "This paper aims to provide an updated perspective on the systemic side effects of AAS abuse."
3. Include a sentence or two on the implications of your findings, and potentially, solutions or recommendations. This will provide a rounded conclusion to the abstract and give an idea of the significance of your work.
2. Section 1:
1. 1.1 Mechanism of action of AAS: You might want to elaborate a bit more on how the modifications of testosterone impact its mechanism of action, the consequences, and why this is significant.
2. 1.2 Prevalence of AAS in the population: The social implications are nicely outlined. However, the impact of these societal pressures could be better highlighted by providing concrete examples or cases.
3. AAS and SARMs: Explain the acronym SARMs (Selective Androgen Receptor Modulators) at its first use.
4. 1.2 Clinical use of AAS: This section ends abruptly. Consider expanding on the role of AAS in treating the mentioned medical conditions and the potential side effects when used in these contexts. How does this compare to the abuse scenarios?
3. Section 2:
This is a well-written section of the manuscript providing an exhaustive explanation of the various known side effects of anabolic-androgenic steroids (AAS). The authors have done an excellent job in citing the references and covering most of the systems affected by AAS.
Here are a few suggestions for improvement:
1. The cardiovascular system section is comprehensive. However, it could benefit from the inclusion of more recent studies (if available) that reinforce or possibly challenge the 1995 hypothetical models proposed by Melchert and Welder.
2. In the musculoskeletal system section, the impact of AAS on tendon strength deserves more elaboration. If possible, add a review of recent studies that might provide more insight or clear out the ambiguity on this topic.
3. In the reproductive system subsection, while the impacts of AAS on males are adequately covered, the female effects could use further development. Especially, the risks or impacts to fertility could be addressed in more detail.
4. In the digestive system section, consider including more on the gastrointestinal effects of AAS, if available, as this might provide a fuller picture of the range of potential impacts.
5. For a general reader, it would be beneficial to include a brief definition or explanation of AAS at the beginning of the section.
6. The authors should consider discussing the psychological effects of AAS use, as these are well-documented and can be severe.
7. The text could benefit from some rephrasing for clarity in places. For instance, the phrase "the increased dimensions of the cavities" could be clarified to specify which cavities are being referred to.
8. Include a section on the endocrine system as AAS has been documented to affect the hypothalamic-pituitary-gonadal axis leading to various effects such as reduced endogenous testosterone production, decreased sperm production, and more.
9. Also, consider integrating the potential interaction of AAS with other substances, if any, and their combined impact on the systems mentioned.
4. Discussion Section:
The discussion section does feel slightly underdeveloped. Given the significant findings presented in the paper, the discussion section could delve deeper into the implications of these results, the limitations of the studies discussed, and potential areas for future research.
5. Contextualization:
The innovative reports are indeed well-detailed and appear comprehensive. However, some context to each subsection, i.e., why each effect (on CNS, psychosocial system, neurodegenerative processes) is crucial to study in the first place, could enrich the narrative further.
6. Conclusion Section:
The text is missing a conclusion section. This is vital for summing up the key points of the research, emphasizing its significance, and providing a takeaway message for the readers.
7. Linking information:
Consider linking information across different sections. For example, how the effects on the central nervous system connect with the psychosocial issues caused by AAS use. Making such connections could enhance the narrative flow of the paper.
8. Substantiation:
The text could benefit from more direct quotations or specific data points from the studies referenced, which could provide a stronger backing for your claims.
9. Future Direction:
In the discussion or conclusion section, it would be beneficial to present a more definitive call to action based on your findings. Suggestions could be for further research, changes in policy, or targeted interventions.
10. Broad Implications:
Broadening the implications of your research beyond the immediate study could provide additional context. For instance, discussing the societal impacts of AAS misuse, how this research might inform policy or public health interventions, etc.
The layout, references, and specific subsections are well-structured. The inclusion of data and results from different studies adds credibility to the claims being made. Great job on this, and good luck with your revisions!
11. General suggestions:
1. Ensure consistent referencing format as per your chosen citation style.
2. Proofread the text for grammatical errors and inconsistencies.
3. Consider including a section on potential methods to mitigate the negative effects of AAS or strategies to prevent misuse.
4. It would also be beneficial to discuss current regulations in place to control AAS misuse and any gaps in these regulations that need to be addressed.
While the subject matter of your paper is deeply engaging and well-researched, there are a few issues regarding the quality of English language used in your manuscript. Here are my comments:
Grammar and Syntax: There are a few instances where grammar and syntax could be improved. While these errors do not obscure the meaning, they might disrupt the flow of reading. For instance, consider changing the sentence structure and maintaining consistent tense usage throughout the manuscript.
Clarity: Some sentences are complex and could benefit from being broken down into simpler sentences. This would enhance readability and comprehension.
Technical Terminology: While the use of technical terms is necessary for a scientific paper, some terms could be better explained or defined, especially for readers who are not deeply familiar with the topic.
Consistency: Be sure to maintain a consistent writing style throughout the document. Some sections are more formal than others, and aligning the tone could help with the overall cohesiveness of the paper.
Jargon: While the technical language is appropriate in academic papers, try to balance it with more accessible language to reach a broader audience.
Proofreading: It would be beneficial to thoroughly proofread the manuscript to eliminate minor typographical errors that could detract from the overall quality of the work.
Overall, the language quality is good, and the message of your research is clear. However, by addressing these issues, you could significantly enhance the readability and accessibility of your paper.
Looking forward to the revised manuscript.
Author Response
Please see the attachment.
We are very grateful for the time that you spent on the review of the manuscript. Kind regards,
Authors

Reviewer 2 Report
For Author
「Our work presents a collection of the latest reports from the literature on groundbreaking discoveries in the study of the side effects of chronic use of AAS.」
1. the author should state his/her "Perspective" after summarizing.
「Testosterone is the most commonly used AAS, and its anabolic ・・・・・・・・・・・・hematological systems and impaired liver function while more recent reports concern neurodegenerative changes.」
2. the section to which this paragraph belongs was unclear (Introduction ? , Abstract ?). The author should have stated the section to which this paragraph belongs (Introduction ? , Abstract ? ). Therefore, this paragraph was not reviewed.
「Receptors for sex hormones are widespread in the brain, especially in such regions as the brainstem, hypothalamus, amygdala, striatum, hippocampus and cerebral cortex [4,6,7].」
3. This review is "Perspective" of cranial nerves only?
「AAS and SARMs (Selective Androgen Receptor Modulators) became a serious public health problem [20, 21]. ・・・・・・・・・・・・however, new research indicates that this statistic is underestimated, and the actual use of doping is even ten times higher [25].」
4. The author should clearly indicate the date/period.
「2.1. Cardiovascular system」
5. After all, please briefly explain the mechanism by which AAS adversely affects the cardiovascular system, using the latest data.
「Muscle damage usually occurs as a result of AAS overuse and independently strong and long-lasting tissue load, which is associated with the risk of injury. 」
6. The reviewer could not understand this sentence.
「The basic mechanism that disturbs active spermatogenesis in these men is the AAS causing ・・・・・by Leydig cells. 」
7. The author should show multiple references.
「Such hormonal changes may lead to azoospermia, oligospermia, testicular atrophy, hypogonadotropic hypogonadism and a reduction in the percentage of morphologically normal spermatozoa」
8. The author should show multiple references.
9. This symptoms are different from those of men using AAS described by the author. Why is that?
「Nonetheless, long-term use may be associated with irreversibility of changes, which has been shown in many studies over the past years.」
10. The reviewer did not understand this sentence.
11. The author should show multiple references.
「alopecia 」
12. Why is there no explanation?
「wide spectrum of sexual dysfunctions」
13. The author should be specific.
「which included as many as 25% of cases.」
14. The reviewer could not understand this sentence.
「Testicular atrophy is associated with reduced secretory activity of Leydig cells. 」
15. The author states that the cause of testicular atrophy is decreased secretory activity. The reviewer did not understand this sentence. The author should elaborate.
「Skin changes associated with AAS appear in the form of papular acne, which is caused by hypertrophy and an increase in the number of sebaceous glands, and thus increased secretion of sebum. [55,56]」
16. what cells respond to AAS?
「Fatigue and depression were common after discontinuation of AAS, as a consequence of the sudden drop in androgenic hormone levels in the blood.」
17. this is not the content of the male reproductive organs. The author should move it to another section.
「Women using AAS are also exposed to a number of ・・・・」
18. in the case of the female genitalia as well as the male genitalia, the author should explain each condition.
「Cases of women using AAS with immature ovarian follicles, hypertrophy of the uterus, clitoris, vagina and mammary gland have been described in the literature.」
19. The author should show multiple references.
「3. Innovative reports on the effects of AAS」
20. the author has separate sections for central nervous system and psychosocial system. The reviewers cannot understand the difference from other SYSTEM.
21. The negative effects of AAS on the human body are important. However, the section "2. Known side effects of AAS" in this review is very cheap. I feel that it needs a major revision.
Author Response

(The authors gave the same response as above.)

Round 2
Reviewer 1 Report
The Authors adequately addressed my questions in the revised manuscript. I accept the present version of the manuscript.
Author Response
Thank you very much for all your comments. We are very pleased that the work has been accepted.
Yours sincerely,
Michał Stojko
Reviewer 2 Report
For authors
「1.1. Mechanism of action of AAS section」
1. The author should indicate the official name of the SAAs.
「3.3. Reproductive system section」
2. The author should indicate the reasons for testicular atrophy due to decreased production of testosterone. decreased Leydig cells? aspermatogenesis?
Author Response
「1.1. Mechanism of action of AAS section」
1. The author should indicate the official name of the SAAs.
Thank you for that. We corrected.
「3.3. Reproductive system section」
2. The author should indicate the reasons for testicular atrophy due to decreased production of testosterone. decreased Leydig cells? aspermatogenesis?
Thank you, we added explanations.
We are very pleased for all your comments.
Yours sincerely,
Michał Stojko
Round 3
Reviewer 2 Report
Congratulations!